# Advantage of Alveolar Ridge Augmentation with Bioactive/Bioresorbable Screws Made of Composites of Unsintered Hydroxyapatite and Poly-L-lactide

**DOI:** 10.3390/ma12223681

**Published:** 2019-11-08

**Authors:** Shintaro Sukegawa, Hotaka Kawai, Keisuke Nakano, Kiyofumi Takabatake, Takahiro Kanno, Hitoshi Nagatsuka, Yoshihiko Furuki

**Affiliations:** 1Department of Oral and Maxillofacial Surgery, Kagawa Prefectural Central Hospital, Takamatsu 765-8557, Japan; furukiy@ma.pikara.ne.jp; 2Department of Oral Pathology and Medicine, Graduate School of Medicine, Dentistry and Pharmaceutical Sciences, Okayama University, Okayama 700-8525, Japan; de18018@s.okayama-u.ac.jp (H.K.); pir19btp@okayama-u.ac.jp (K.N.); gmd422094@s.okayama-u.ac.jp (K.T.); jin@okayama-u.ac.jp (H.N.); 3Department of Oral and Maxillofacial Surgery, Shimane University Faculty of Medicine, Shimane 693-8501, Japan; tkanno@med.shimane-u.ac.jp

**Keywords:** poly-L-lactide, uncalcined and unsintered hydroxyapatite, biocompatibility, osteoconductivity, mesenchymal stem cell

## Abstract

We studied human bone healing characteristics and the histological osteogenic environment by using devices made of a composite of uncalcined and unsintered hydroxyapatite (u-HA) and poly-L-lactide (PLLA). In eight cases of fixation, we used u-HA/PLLA screws for maxillary alveolar ridge augmentation, for which mandibular cortical bone block was used in preimplantation surgery. Five appropriate samples with screws were evaluated histologically and immunohistochemically for runt-related transcription factor 2 (RUNX2), transcription factor Sp7 (Osterix), and leptin receptor (LepR). In all cases, histological evaluation revealed that bone components had completely surrounded the u-HA/PLLA screws, and the bone was connected directly to the biomaterial. Inflammatory cells did not invade the space between the bone and the u-HA/PLLA screw. Immunohistochemical evaluation revealed that many cells were positive for RUNX2 or Osterix, which are markers for osteoblast and osteoprogenitor cells, in the tissues surrounding u-HA/PLLA. In addition, many bone marrow–derived mesenchymal stem cells were notably positive for both LepR and RUNX2. The u-HA/PLLA material showed excellent bioactive osteoconductivity and a highly biocompatibility with bone directly attached. In addition, our findings suggest that many bone marrow–derived mesenchymal stem cells and mature osteoblast are present in the osteogenic environment created with u-HA/PLLA screws and that this environment is suitable for osteogenesis.

## 1. Introduction

Titanium fixation devices have been used widely as a standard for maxillofacial surgery because they are easy to operate and relatively inexpensive; however, plate removal may be necessary, and various complications can be caused by the metal [1]. Therefore, bioresorbable fixation devices made of synthetic polymers are currently used widely as an alternative material for internal fixation. An ideal bioresorbable osteosynthesis device should have the proper modulus and high strength, retain that strength as long as bone healing requires support, and be safely absorbed and disassembled without a foreign body reaction that delays the bone-healing process.

Bioabsorbable fixation devices made of high-strength uncalcined and unsintered hydroxyapatite (u-HA) and poly-L-lactide (PLLA) composites have been developed to solve the mechanical and biological problems of life-long implants [2]. Currently, Super FIXSORB MX^®^ (Teijin Medical Technologies Co., Ltd. Osaka, Japan), also known as OSTEOTRANS MX, can be used as a commercially available u-HA/PLLA osteosynthesis bioresorbable device. This bioresorbable device, which consists of u-HA and PLLA, is manufactured by a compression molding reinforcement process and a forging process incorporating machining. Because of its composition and the special manufacturing process, this device has higher mechanical strength and bioactivity [2,3,4,5]. The bioactivity of bioresorbable plates is a major advantage, and their bone conduction and bone-binding ability [6,7], complete long-term replacement of the human bone [8], and biocompatibility [6,7,8] have been reported. In addition, we have previously reported the presence of osteoblast differentiation markers in the environment surrounding u-HA/PLLA materials [7], which has already shown that u-HA/PLLA materials are bioactive materials with excellent bone regeneration ability.

However, the bone-healing properties of this device and the histological environment for bone healing remain unclear. In this study, we investigated bone-healing characteristics and the histological environment for u-HA/PLLA composite devices to understand the in vivo environment when this device is used in maxillofacial clinical treatment.

## 2. Materials and Methods

### 2.1. Preparation of Uncalcined and Unsintered Hydroxyapatite/Poly-L-lactide Composite Screws

In this study, we used the Super FIXSORB MX® screw (Teijin Medical Technologies Co., Ltd. Osaka, Japan), comprising a forged composite of u-HA/PLLA (containing 30 weight fractions of raw uncalcined, unsintered HA particles in composites). The screws have a diameter of 2.0 mm and a length of 8–12 mm; u-HA particle size ranges from 0.2 to 20 μm (average size, 3–5 μm); the ratio of HA weight to PLLA weight is 30/70; the ratio of calcium to phosphorus is 1.69 (moles); and CO_3_^2−^ level is 3.8 (percentage of moles). The composite material used in this study was the same as that reported in the past [2].

### 2.2. Subjects

This study included eight consecutive patients (two men and six women; age range, 33–59 years) who needed maxillary alveolar ridge augmentation as preimplantation surgery because their residual bone width was <4 mm; informed consent to participate in the study was obtained from all the patients. All operations were performed by a single oral and maxillofacial surgeon (Shintaro Sukegawa) from April 2018 to March 2019 in the Department of Oral and Maxillofacial Surgery at Kagawa Prefectural Central Hospital, Takamatsu, Kagawa, Japan. The cases of this study are shown in Table 1.

### 2.3. Surgical Bone Augmentation Procedure

In the surgical operation, the amount of material necessary for bone augmentation was collected from the buccal cortical bone block of the mandibular ramus. The cortical bone block was fixed to the recipient site by using u-HA/PLLA screws. The screw fixing method consisted of the following steps: (1) drilling to form a bone hole, (2) forming a tap with a screw tap, and (3) insertion of u-HA/PLLA screws into the holes formed by self-tapping. The screw insertion torque was 5 N. The number of screws used was selected to obtain stable fixation of the bone block (Figure 1a).

### 2.4. Sample Collection

After approximately 6 months, to allow for bone healing, dental implant placement was planned with the use of computed tomography. (Figure 1b). At the time of implant placement, specimens were collected with a 2.0 mm diameter trephine bar (ACE Surgical Supply Company, Inc., Brockton, MA, USA) (Figure 1c). All procedures were performed by the same expert surgeon (Shintaro Sukegawa) at the same institution. This study was approved by the Ethics Committee of the Kagawa Prefectural Central Hospital (Approval No. 879).

### 2.5. Preparation for Histological Evaluation

All samples were immediately fixed in 4% paraformaldehyde for 12 h and then decalcified in 10% ethylenediaminetetraacetic acid at 4 °C for 14 days. Samples were dehydrated with a graded series, soaked in xylol several times, and embedded in paraffin. Thin serial sections were made from samples embedded in paraffin. The sections were used for hematoxylin–eosin staining and immunohistochemical study.

### 2.6. Immunohistochemistry

The expressions of runt-related transcription factor 2 (RUNX2), transcription factor Sp7 (Osterix), and leptin receptor (LepR) were evaluated in an immunohistochemical study. The prepared sample paraffin-embedded block was sectioned in thicknesses of 3 μm. These sections were deparaffinized in xylene for 15 min and rehydrated in graded ethanol solution. To prevent endogenous peroxidase activity, the sections were incubated in 0.3% H_2_O_2_ and methanol for 30 min. Antigen retrieval was achieved by heat treatment with 10-mM citrate buffer solution at a pH of 9.0. After treatment with normal serum, the sections were incubated with the primary antibodies for RUNX2 (Abcam plc., Cambridgesshire, ab23981, UK, dilution of 1:500), Osterix (Abcam plc., Cambridgesshire, ab22552, UK, dilution of 1:100), and LepR (Proteintech. 20966-1, USA, dilution of 1:50) at 4 °C overnight. To tag the primary antibody, EnVision peroxidase detecting reagent (Dako, Carpinteria, CA, USA) was applied. We identified the immunoreactive site by using the avidin–biotin complex method (Vector Laboratories, Burlingame, CA, USA). Detection was performed with 3,3′-Diaminobenzidine (DAB), and the staining results were observed with an optical microscope.

For double-fluorescent Immunohistochemistry (IHC), the abovementioned LepR and RUNX2 antibodies were used as primary antibodies. Antibodies were diluted with Can Get Signal (Toyobo, Osaka, Japan). Anti-mouse IgG Alexa Fluor 488 (Life technologies, Waltham, MA, USA) and anti-rabbit IgG Alexa Fluor 568 (Carlsbad, CA, USA) were used as secondary antibodies at a dilution of 1:200. After the reactions, the specimens were stained with 1 mg/mL of DAPI (Dojindo Laboratories, Kumamoto, Japan). The staining results were observed with a fluorescence microscope.

## 3. Results

### 3.1. Clinical Evaluation

Six months after anterior maxillary alveolar bone augmentation of preimplantation surgery, bone width in all patients was sufficient for placing the dental implant. The transplanted cortical bone block was fully engrafted in all patients. No complications were observed in any of the patients after dental implant placement, and all results with the final prosthesis set were satisfactory. We used the trephine bar to obtain eight specimens from the bone-constructed area with u-HA/PLLA screws at the time of implant placement. Of these specimens, five in which the implant was placed in the same location as the u-HA/PLLA screw were examined histologically.

### 3.2. Histopathological Evaluations

Eight specimens were examined for resected material by using hematoxylin–eosin staining. In five specimens, we were able to observe both screws and the surrounding bone. In 2 cases (Cases 1 and 2), bone components had completely surrounded the u-HA/PLLA screws (Figure 2a,c). The high magnification field revealed that the bone was directly connected to the biomaterial and that inflammatory cells did not invade the space between the bone and the u-HA/PLLA screw. There was no cellular infiltration between the bone and the u-HA/PLLA screw, and the materials were completely continuous. No foreign body reaction was induced around the u-HA/PLLA screws (Figure 2a,b,d,e). These findings indicate that u-HA/PLLA screw has high bone compatibility.

In other cases (Cases 3–5) fibrous tissue was observed surrounding the u-HA/PLLA. This histological character of the fibrous tissue was uniform, and there was no inflammation and bleeding. Foreign body giant cells were not present in stromal tissue (Figure 3b,c,e,f,h,i). Furthermore, this fibrous tissue contained bone tissue and was continuous (Figure 3c,f,i). These findings are different from those reported in other two cases (Cases 1 and 2), but findings of both these cases suggested that u-HA/PLLA screws are highly biocompatible. In addition, the results of all the three cases (Cases 3–5) indicate the potential for bone making ability around the u-HA/PLLA.

### 3.3. Immunohistochemical Evaluations

To investigate the characteristics of fibrous tissue, we performed immunostaining. Because of absence of inflammation and existing new bone, we selected the marker for preosteoblast (RUNX2, Osterix) and mesenchymal stem cell marker (LepR). In the fibrous tissue, RUNX2- or Osterix-positive cells were observed; these cells were spindle shape and scattered (Figure 4a,b). LepR-positive cells were also observed in fibrous tissue and their shape was similar to RUNX2- or Osterix-positive cells (Figure 4c). It was confirmed that a number of RUNX2-positive cells expressed LepR (Figure 4d–f).

## 4. Discussion

In this study, the use of u-HA/PLLA screws in maxillofacial bone augmentation with bone block as preimplantation surgery yielded reliable results. The u-HA/PLLA screws were highly biocompatible with the bone. Additionally, in the tissues surrounding the u-HA/PLLA screw, many cells were positive for RUNX2 and Osterix, which are markers for osteoprogenitor cells. Furthermore, several cells positive for both LepR and RUNX2.

Bioresorbable osteosynthesis devices made of a variety of synthetic polymers such as polyglycolide, PLLA, polydioxanone, or glycolide–lactide copolymers have been developed [9]. However, clinical studies of conventional resorbable devices have demonstrated various complications, including mechanical weakness, osteolytic changes around the device [10,11], and degradation of the tissue [12]. To overcome these limitations, composite materials comprising bioactive ceramics as fillers and PLLA as matrices have been developed. These composites are intended to provide both biocompatibility with the bone and desirable mechanical properties, including polymer ductility and stiffness equal to or better than that of cortical bone. The composite material consisting of u-HA as a bioactive ceramic and PLLA as a matrix has great advantages. The u-HA/PLLA composite material has an initial bending strength of 280 MPa, higher than that of human cortical bone (120–210 MPa), and an elastic modulus of 12 GPa, which is mechanically stronger than any other bioactive ceramic/polymer composites available to date [13]. The u-HA/PLLA composite material has not only high strength but also bioactivity advantages [14].

Surprisingly, the u-HA/PLLA screw was bound directly to the human bone in all specimens in this study in which the screw was detected. Hydroxyapatite was reported to have formed directly on the surface of the composite material after immersion in a simulated body fluid in in vitro research [2]. When cells touch the surface of material, they usually attach, adhere, and spread. The first stage of this interaction between cell and material depends on the characteristics of the material surface to determine the behavior of the cell upon contact with the material. Osteoblasts have been found to preferentially attach to HA particles via filopodia, demonstrating that HA provides a favorable anchoring site for human osteoblast adhesion [15]. Hydroxyapatite on the surface of u-HA/PLLA screw may play an important role in direct bonding with human bone. This osteoconductive feature is a major advantage of this material.

Of histological importance is the lack of foreign body reaction around the u-HA/PLLA material. In this study, new bone is formed through direct contact with material and human bone. For regenerative bone formation, the implant material must elicit minimal inflammatory reaction. In the past, we quantitatively evaluated the presence of CD68, considered a marker for macrophages, around u-HA/PLLA screws. CD68 was rarely observed around the u-HA/PLLA material [7]. In this study, inflammatory cells and giant cells against foreign bodies were not observed in any of the specimens examined, indicating high biocompatibility of the u-HA/PLLA material.

There are two type of pathways in the bone-healing process: intramembranous ossification and endochondral ossification [16]. Intramembrane ossification is a process formed by mesenchymal cells that condense to become functional osteoblasts without cartilage formation. Endochondral ossification, in contrast, is a process of replacing a cartilage template composed of chondrocytes differentiated from mesenchymal stem cells with bone containing osteoblasts and osteoclasts. These cells are functionally responsible for bone formation that directs the deposition and calcification of bone matrix. Osteoblasts are induced through the expression of osteoblast-specific transcription factor RUNX2 and Osterix from immature mesenchymal stromal cells [17,18]. In our immunostaining evaluation, RUNX2- or Osterix-positive cells were localized in the tissue surrounding u-HA/PLLA material. These cells were present in connective tissue and did not secrete bone matrix. Numerous preosteoblasts were present in the surrounding stroma of u-HA/PLLA, which suggests that this microenvironment might form bone tissue.

In recent studies, bone marrow derived mesenchymal stem cells have been identified as LepR-positive cells in cell lineage analysis [19]. Yang et al. reported that approximately 60% of LepR-positive cells expressed RUNX2 and that among the LepR-positive cells, the RUNX2-positive subpopulation had higher stem cell capacity than did the RUNX2-negative subpopulation [19]. Furthermore, cells that were both LepR and RUNX2 positive showed pluripotency in an in vitro culture system [20]. The findings by Yang et al. suggest that LepR- and RUNX2-positive cells are located upstream of the differentiation tree of bone marrow mesenchymal cells. Our results also showed that LepR- and RUNX2- double positive cells were expressed in the environment surrounding u-HA/PLLA screws. It has already been shown that u-HA/PLLA materials are bioactive materials with excellent bone regeneration ability. However, the bone formation environment has not been elucidated. This study is the first report demonstrating the surrounding environment with LepR- and RUNX2-positive cells following the placement of u-HA/PLLA screws. The results of this study suggested that the bone formation environment performed with u-HA/PLLA screws may enable the expression of bone marrow–derived mesenchymal cells, which is a good environment for bone formation. Moreover, these findings indicate that the screw induce recruitment of bone marrow derived mesenchymal stem cells (RUNX2/LeptinR double positive).

A limitation of this study was that it was not possible to evaluate the tissue changes over long term for materials implanted in the human body. We evaluated the timing of dental implant placement after bone formation and samples collected at the same time as the dental implant placement. Because resorbable osteosynthesis cannot be removed without complications, investigation through animal experiments may enable evaluation at other times. On the contrary, to the best of our knowledge, the results of this study will help elucidate the bone-healing properties of, and the histological environment created by, u-HA/PLLA bioresorbable material in human maxillofacial bone. This is interesting and important with regard to the in vivo response to u-HA/PLLA bioresorbable materials.

## 5. Conclusions

The u-HA/PLLA screws demonstrated excellent bioactive osteoconductivity and high biocompatibility with maxillofacial bone in this study. In the tissues surrounding the u-HA/PLLA material, many cells were positive for RUNX2 and Osterix, the markers for osteoprogenitor cells. In addition, several bone marrow–derived mesenchymal stem cells were positive for both LepR and RUNX2. These results suggest that the bone formation stimulated by u-HA/PLLA screws may provide a good environment for bone regenerative formation in maxillofacial surgery.

## Figures and Tables

**Figure 1 materials-12-03681-f001:**
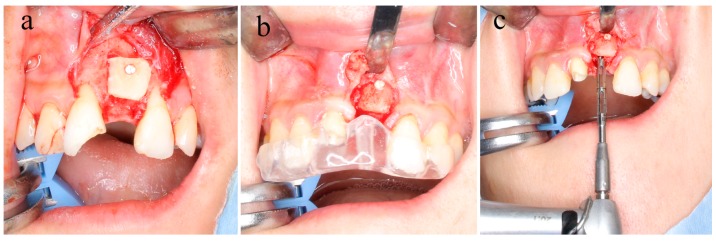
(**a**) In the surgical operation, the amount necessary for bone augmentation was collected from the buccal cortical bone block of the mandibular ramus, and the buccal cortical bone block was fixed to the recipient site with uncalcined and unsintered hydroxyapatite/poly-L-lactide (u-HA/PLLA) screws. (**b**,**c**) Six months later, dental implantation surgery was planned. The position of the u-HA/PLLA screw was confirmed. At the time of implant placement, specimens were collected using a 2.0 mm diameter trephine bar.

**Figure 2 materials-12-03681-f002:**
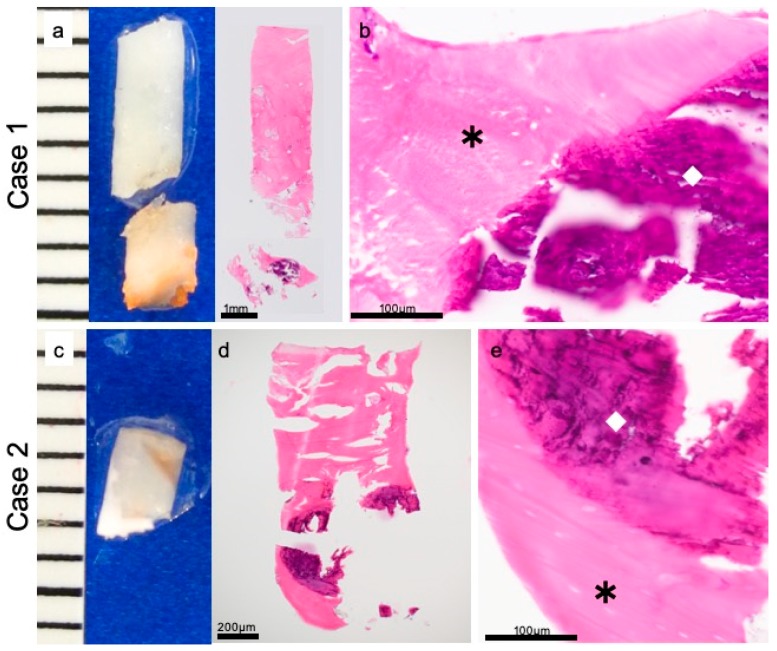
Direct contact of material and bone. Case 1: (**a**) macro findings and hematoxylin-eosin staining. (**b**) Hematoxylin–eosin staining. Case 2: (**c**) macro findings. (**d**,**e**) Hematoxylin–eosin staining. In both cases, the uncalcined and unsintered hydroxyapatite/poly-L-lactide (u-HA/PLLA) screw (◇) and bone (*) are in direct contact, with no connective tissue interposed between them.

**Figure 3 materials-12-03681-f003:**
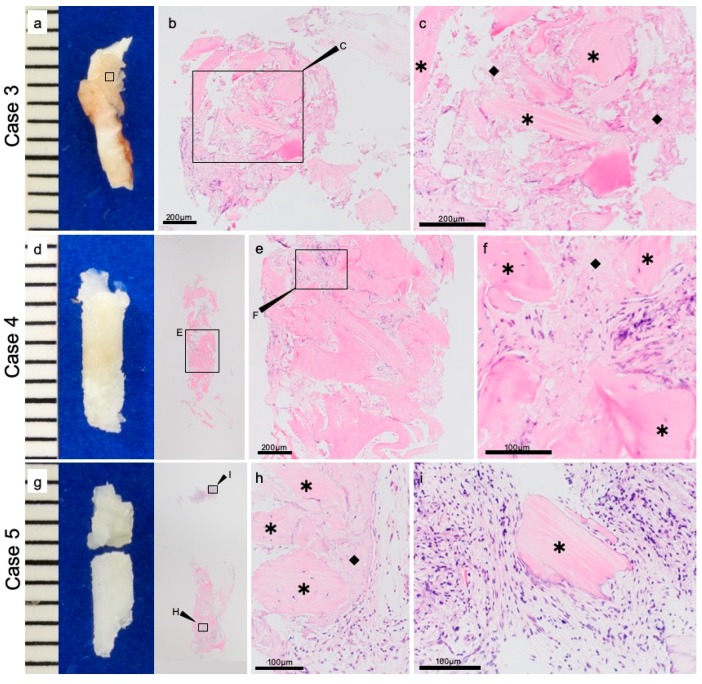
The fibrous tissue around the screw. Case 3: (**a**) macro findings. (**b**,**c**) Hematoxylin–eosin staining. Case 4: (**d**) macro findings. (**e**,**f**) Hematoxylin–eosin staining. Case 5: (**g**) macro findings. (**h**,**i**) Hematoxylin–eosin staining. In the tissue surrounding, the diamond (◆) indicate the u-HA/PLLA screw, and the star (*) indicate the bone.

**Figure 4 materials-12-03681-f004:**
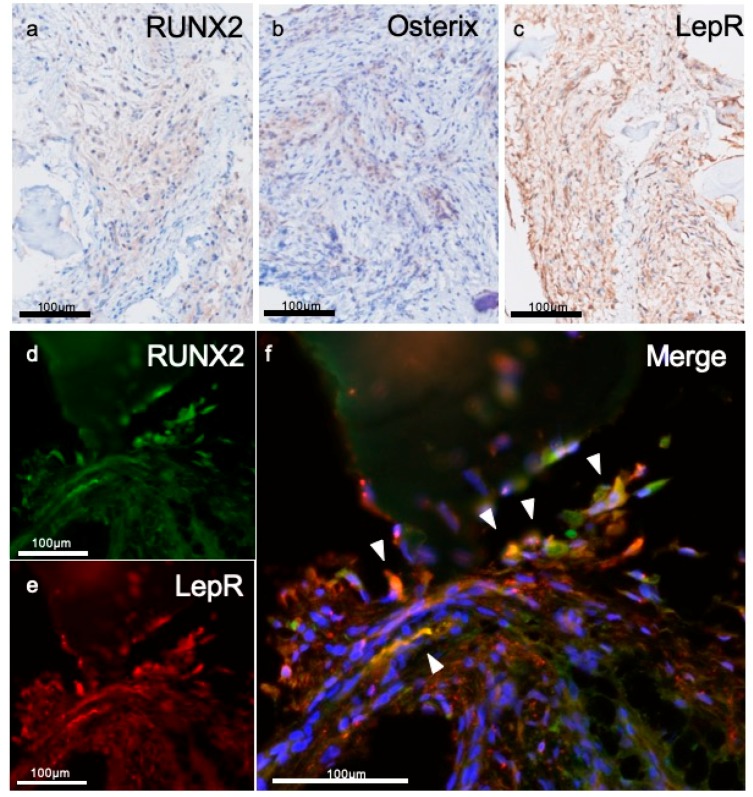
The characteristics of fibrous tissue around the screw. Immunohistochemistry for (**a**) RUNX2, (**b**) Osterix, and (**c**) leptin receptor (LepR). Spindle cells are positive for each marker. Double florescent IHC for fibrous tissue surrounding screw. Several cells are positive for (**d**) RUNX2 or (**e**) LepR, and (**f**) some cells were positive for both markers (arrow head).

**Table 1 materials-12-03681-t001:** Details of patients and u-HA/PLLA screws used.

Patient Number	Sex (Male/Female)	Age (Years)	Screw Length (mm)	Number of Screws	Presence or Absence of u-HA/PLLA Screws in the Specimen	Period from the Screw Placement to Evaluation (Day)
1	Female	44	8.0	2	Presence	246
2	Female	57	8.0	2	Presence	209
3	Female	59	8.0	2	Absence	219
4	Female	55	12.0	3	Presence	203
5	Male	55	12.0	1	Absence	223
6	Female	33	12.0	1	Presence	209
7	Male	50	8.0	2	Presence	226
8	Female	58	8.0	2	Absence	212

u-HA/PLLA, uncalcined and unsintered hydroxyapatite/poly-L-lactide.

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
