# Peer review of "Advantage of Alveolar Ridge Augmentation with Bioactive/Bioresorbable Screws Made of Composites of Unsintered Hydroxyapatite and Poly-L-lactide"

_materials, 2019, doi:10.3390/ma12223681_

Round 1

Reviewer 1 Report

Manuscript ID: materials-616990

Title: Advantage of Alveolar Ridge Augmentation with Bioactive/Bio-resorbable Screws Made of Composites of Unsintered Hydroxyapatite and Poly-L-Lactide

Topic and Findings: The authors report the “investigation of bone-healing characteristics and the histological environment for u-HA/PLLA composite devices to understand the in vivo environment when this device is used in maxillofacial clinical therapy treatment.”

The topic is of interest, minor revisions are required.

Introduction:

In particular, the description of the status quo of synthetic biomaterials is very weak. There are numerous studies on PLLA-based composites. The entire introduction cites 8 references in total. Please explain. In addition, please specify the differences between this manuscript and the published paper (reference No. 7):

Sukegawa, S.; Kawai, H.; Nakano, K.; Kanno, T.; Takabatake, K.; Nagatsuka, H.; Furuki, Y. Feasible 287 advantage of bioactive/bioresorbable devices made of forged composites of hydroxyapatite particles and 288 poly-L-lactide in alveolar bone augmentation: A preliminary study. Int. J. Med. Sci. 2019, 16, 311–317.

Please explain and specify more in detail the novelty of the submitted study in comparison to studies performed and already published:

Materials and Methods:

Lines 61/62: “The ratio of HA weight to PLLA weight is 30/70; the ratio of calcium to phosphorus is 1.69 (moles); and CO32− level is 3.8 (percentage of moles).”

What methods have been used to characterize the final composite.

Specify the statistics for the performed experiments.

Results and Discussion:

Lines 195/196: “The u-HA/PLLA 195 composite material has not only high strength but also bioactivity advantages [14].”

Please explain/specify the novelty of the submitted study in comparison to studies performed and previously published.

Dong, QN.; Kanno, T.: Bai, Y.; Sha, J.; Hideshima, K. Bone Regeneration Potential of Uncalcined and 305 Unsintered Hydroxyapatite/Poly l-lactide Bioactive/Osteoconductive Sheet Used for Maxillofacial 306 Reconstructive Surgery: An In Vivo Study. Materials (Basel). 2019, 11, 12.

Author Response

Points/Contents and Responses to the Reviewer’s Suggestions

Thank you very much for your valuable comments and kind acceptance. We have incorporated all the reviewer’s comments and suggestions into our manuscript, as red letters in the revised manuscript. We would like to say thank you for the reviewer’s suggestions, which were very helpful to further, improve this manuscript.

The lists of suggestions from reviewer with answers.

Comments: Reviewer 1

Topic and Findings: The authors report the “investigation of bone-healing characteristics and the histological environment for u-HA/PLLA composite devices to understand the in vivo environment when this device is used in maxillofacial clinical therapy treatment.”

The topic is of interest, minor revisions are required.

Reviewer comment: In particular, the description of the status quo of synthetic biomaterials is very weak. There are numerous studies on PLLA-based composites. The entire introduction cites 8 references in total. Please explain. In addition, please specify the differences between this manuscript and the published paper (reference No. 7):

Author response: We thank the reviewer for these helpful comments. Reference No. 7 reports the presence of osteoblast differentiation markers in the environment surrounding u-HA / PLLA material. In this report, we report the upstream mechanism of osteoblasts in the environment surrounding u-HA / PLLA materials. And more, we added the description of the status quo of synthetic biomaterials of u-HA/PLLA

Reviewer comment: Lines 61/62: “The ratio of HA weight to PLLA weight is 30/70; the ratio of calcium to phosphorus is 1.69 (moles); and CO32− level is 3.8 (percentage of moles).”

What methods have been used to characterize the final composite.

Specify the statistics for the performed experiments.

Author response: We thank the reviewer for these helpful comments.  The screws used in this study were the same as those reported in the past, and we referred to them. Reference is No2.

Reviewer comment: Lines 195/196: “The u-HA/PLLA composite material has not only high strength but also bioactivity advantages [14].”

Please explain/specify the novelty of the submitted study in comparison to studies performed and previously published.

Author response: We thank the reviewer for these helpful comments. It has already been shown that u-HA / PLLA materials are bioactive materials with excellent bone regeneration ability. However, the detailed bone formation environment has not been elucidated. This study reports for the first time a partial mechanism of bone formation that contributed to the excellent bone regeneration ability of this material. (Line239-242)

Reviewer 2 Report

In my humble opinion, this well-written manuscript may be published as is. No corrections are necessary.

Author Response

Points/Contents and Responses to the Reviewer’s Suggestions

Thank you very much for your valuable comments and kind acceptance. We have incorporated all the reviewer’s comments and suggestions into our manuscript, as red letters in the revised manuscript. We would like to say thank you for the reviewer’s suggestions, which were very helpful to further, improve this manuscript.

The lists of suggestions from reviewer with answers.

Comments:

Reviewer2:

In my humble opinion, this well-written manuscript may be published as is. No corrections are necessary.

Author response: Thank you so much for your very pleasant evaluation comment for our research in terms of study of historical evaluation in the osteogenic environment created with u-HA/PLLA screws.

Reviewer 3 Report

Review for materials-616990-peer-review-v1

General Comments: The manuscript is nicely written and an exciting use of this implant (screws). I have one major comment, it is about the use of “osteoprogenitor” and “mesenchymal stem cells”. True, Runx2+ and LepR+ cells are osteoprogenitor cells, likely mesenchymal stem cells. However, I would refer to the Runx2+ and Osx+ population as osteoblasts. While they are not as mature as Runx2+ OCN+ terminally differentiated osteoblasts that are essentially “osteocytes” when bedding in ECM, Osx expression peaks around day 14 of the 28-30 day differentiation cascade from day 0 pre-osteoblasts to day 28-30 terminally differentiated osteoblasts. For this reason, I feel that Runx2+ Osx+ cells are in fact osteoblasts, and not “osteoprogenitor cells”

More specific comments and text editing/text corrections and listed below by line number.

More Specific Comments:

Title – None

Abstract

Line 14 – Change “bone-healing” to “bone healing” Line 14-15 – Change “histological osteogenic environment” to “osteogenic environment by histology” Line 24 – See general comments above Line 28 – Include that in addition to MSCs, more mature osteoblasts are present also

Introduction

Line 55 – “maxillofacial clinical therapy treatment” needs to be reworded

Materials and Methods – None

Results

Line 138 – For “boundary”, select a better phrase such as “inflammatory membrane” “inflammatory cells” or “cellular infiltrate” since you are referring to a cellular process. Line 149 – Change “Case 3-5” to “Cases 3-5”

Discussion

Line 209 – For “no inflammatory reaction” it should be “minimal inflammatory reaction” Line 211-213 – For the part about inflammatory and giant cells, expand on this more in the Results section. Line 221 – Osteoblasts need both Runx2 AND Osx. Line 236-237 – Starting with “expression of bone”, reword this passage. Line 238 – For “mobilization” should it also be “recruitment”?

Conclusions - See general comments above about cell nomenclature

Figures, Tables, and Legends

Table 1 – Last 2 columns – This is not referenced in the text. Figure 2 legend line 147 – See above comment about “boundary” when you are referring to “connective tissue” here

Author Response

Points/Contents and Responses to the Reviewer’s Suggestions

Thank you very much for your valuable comments and kind acceptance. We have incorporated all the reviewer’s comments and suggestions into our manuscript, as red letters in the revised manuscript. We would like to say thank you for the reviewer’s suggestions, which were very helpful to further, improve this manuscript.

The lists of suggestions from reviewer with answers.

Comments: Reviewer 3

General Comments: The manuscript is nicely written and an exciting use of this implant (screws). I have one major comment, it is about the use of “osteoprogenitor” and “mesenchymal stem cells”. True, Runx2+ and LepR+ cells are osteoprogenitor cells, likely mesenchymal stem cells. However, I would refer to the Runx2+ and Osx+ population as osteoblasts. While they are not as mature as Runx2+ OCN+ terminally differentiated osteoblasts that are essentially “osteocytes” when bedding in ECM, Osx expression peaks around day 14 of the 28-30 day differentiation cascade from day 0 pre-osteoblasts to day 28-30 terminally differentiated osteoblasts. For this reason, I feel that Runx2+ Osx+ cells are in fact osteoblasts, and not “osteoprogenitor cells”

More specific comments and text editing/text corrections and listed below by line number.

Author response: Thank you for valuable commenting. We agree to your opinion, Runx2 + Osx + cells are osteoblasts in terms of cell differentiation marker research. However, in this study, Runx2 + or Osx + cells were placed in the stromal area. Osteoblasts are present on the bone surface. Because Runx2 + or Osx + cells were observed in the stromal fibrous tissue, the cells were considered osteoprogenitors rather than osteoblasts. We think a part of Runx2 positive cells are BM derived mesenchymal stem cells.

More Specific Comments:

Title – None

Abstract

Line 14 – Change “bone-healing” to “bone healing” Line 14-15 – Change “histological osteogenic environment” to “osteogenic environment by histology” Line 24 – See general comments above Line 28 – Include that in addition to MSCs, more mature osteoblasts are present also

Author response: We thank the reviewer for these helpful comments. As you pointed out, we corrected each part.

Introduction

Line 55 – “maxillofacial clinical therapy treatment” needs to be reworded

Author response: We thank the reviewer for these helpful comments. We reworded to “maxillofacial clinical treatment”.

Materials and Methods – None

Results

Line 138 – For “boundary”, select a better phrase such as “inflammatory membrane” “inflammatory cells” or “cellular infiltrate” since you are referring to a cellular process. Line 149 – Change “Case 3-5” to “Cases 3-5”

Author response: We thank the reviewer for these helpful comments. As you pointed out, we corrected each part.

Discussion

Line 209 – For “no inflammatory reaction” it should be “minimal inflammatory reaction” Line 211-213 – For the part about inflammatory and giant cells, expand on this more in the Results section. Line 221 – Osteoblasts need both Runx2 AND Osx. Line 236-237 – Starting with “expression of bone”, reword this passage. Line 238 – For “mobilization” should it also be “recruitment”?

Author response: We thank the reviewer for these helpful comments. As you pointed out, we corrected each part.

Conclusions - See general comments above about cell nomenclature

Figures, Tables, and Legends

Table 1 – Last 2 columns – This is not referenced in the text. Figure 2 legend line 147 – See above comment about “boundary” when you are referring to “connective tissue” here 

Author response: We thank the reviewer for these helpful comments. As you pointed out, we corrected each part.

Round 2

Reviewer 3 Report

The authors have nicely addressed all comments. No further edits or comments!